# Estimating Training Data Influence
# by Tracing Gradient Descent

**Garima**[*]
Google
pruthi@google.com

**Frederick Liu**[*]
Google
frederickliu@google.com

**Satyen Kale**
Google
satyenkale@google.com

**Mukund Sundararajan** [†]
Google
mukunds@google.com

## Abstract

We introduce a method called `TracIn` that computes the influence of a training example on a prediction made by the model. The idea is to trace how the loss on the test point changes during the training process whenever the training example of interest was utilized. We provide a scalable implementation of `TracIn` via: (a) a first-order gradient approximation to the exact computation, (b) saved checkpoints of standard training procedures, and (c) cherry-picking layers of a deep neural network. In contrast with previously proposed methods, `TracIn` is *simple* to implement; all it needs is the ability to work with gradients, checkpoints, and loss functions. The method is *general*. It applies to any machine learning model trained using stochastic gradient descent or a variant of it, agnostic of architecture, domain and task. We expect the method to be widely useful within processes that study and improve training data. Code is available at [1].

## 1 Motivation

Deep learning has been used to solve a variety of real-world problems. A common form of machine learning is supervised learning, where the model is trained on *labelled* data. Controlling the training data input to the model is one of the main quality knobs to improve the quality of the deep learning model. For instance, such a technique could be used to identify and fix mislabelled data using the workflow described in Section 4.1. Our main motivation is to identify practical techniques to improve the analysis of the training data. Specifically, we study the problem of *identifying the **influence of training examples on the prediction of a test example***. We propose a method called `TracIn` for computing this influence, provide a scalable implementation, and evaluate the method experimentally.

## 2 Related Work

[2, 3] tackle influential training examples in the context of deep learning. We discuss these methods in detail in Section 4.4.

There are related notions of influence used to explain deep learning models that differ in either the target of the explanation or the choice of influencer or both. For instance, [4, 5, 6] identify the influence of features on an individual prediction. [7, 8] identify the influence of features on the

---

[*]Equal contribution.

[†]Corresponding author.

overall accuracy (loss) of the model. [9, 10] identify the influence of training examples on the overall accuracy of the model. [11], a technique closely related to `TracIn`, identifies the influence of training examples on the overall loss by tracing the training process while `TracIn` identifies the influence on the loss of a test point. The key trick in [11] is to use a certain hessian of the model parameters to trace the influence of a training point through minibatches in which it is *absent*. This trick is also potentially useful in implementing idealized version of `TracIn`. However, idealized `TracIn` requires the test/inference point to be known at training time and is therefore impractical, making the trick less relevant to the problem we study. `TracInCP`, a practical implementation, leverages checkpoints to replay the training process. Checkpoint ensembling is a widely used technique in machine translation [12], semi-supervised learning [13] and knowledge distillation [14] which provide intuition on why `TracIn` performs better than other methods.

## 3 The Method

In this section we define `TracIn`. `TracIn` is inspired by the *fundamental theorem of calculus*. The fundamental theorem of calculus decomposes the difference between a function at two points using the gradients along the path between the two points. Analogously, `TracIn` decomposes the difference between the loss of the test point at the end of training versus at the beginning of training along the path taken by the training process.[3]

We start with an idealized definition to clarify the idea, but this definition will be impractical because it would require that the test examples (the ones to be explained) to be specified at training time. We will then develop practical approximations that resolve this constraint.

### 3.1 Idealized Notion of Influence

Let $Z$ represent the space of examples, and we represent training or test examples in $Z$ by the notation $z, z'$ etc. We train predictors parameterized by a weight vector $w \in \mathbb{R}^p$. We measure the performance of a predictor via a loss function $\ell : \mathbb{R}^p \times Z \to \mathbb{R}$; thus, the loss of a predictor parameterized by $w$ on an example $z$ is given by $\ell(w, z)$.

Given a set of $n$ training points $S = \{z_1, z_2, \ldots, z_n \in Z\}$, we train the predictor by finding parameters $w$ that minimize the training loss $\sum_{i=1}^n \ell(w, z_i)$, via an iterative optimization procedure (such as stochastic gradient descent) which utilizes *one* training example $z_t \in S$ in iteration $t$, updating the parameter vector from $w_t$ to $w_{t+1}$. Then the idealized notion of influence of a particular **training example** $z \in S$ on a given **test example**[4] $z' \in Z$ is defined as the total reduction in loss on the test example $z'$ that is induced by the training process whenever the training example $z$ is utilized, i.e. $\texttt{TracInIdeal}(z, z') = \sum_{t:\ z_t=z} \ell(w_t, z') - \ell(w_{t+1}, z')$

Recall that `TracIn` was inspired by the fundamental theorem of calculus, which has the property that the integration of the gradients of a function between two points is equal to the difference between function values between the two points. Analogously, idealized influence has the appealing property that the sum of the influences of all training examples on a fixed test point $z'$ is exactly the total reduction in loss on $z'$ in the training process:

**Lemma 3.1** *Suppose the initial parameter vector before starting the training process is $w_0$, and the final parameter vector is $w_T$. Then $\sum_{i=1}^n \texttt{TracInIdeal}(z_i, z') = \ell(w_0, z') - \ell(w_T, z')$*

Our treatment above assumes that the iterative optimization technique operates on one training example at a time. Practical gradient descent algorithms almost always operate with a group of training examples, i.e., a *minibatch*. We cannot extend the definition of idealized influence to this setting, because there is no obvious way to redistribute the loss change across members of the minibatch. In Section 3.2, we will define an approximate version for minibatches.

**Remark 3.2 (Proponents and Opponents)** *We will term training examples that have a positive value of influence score as **proponents**, because they serve to reduce loss, and examples that have*

*a negative value of influence score as **opponents**, because they increase loss. In [2], proponents are called 'helpful' examples, and opponents called 'harmful' examples. We chose more neutral terms to make the discussions around mislabelled test examples more natural. [3] uses the terms 'excitory' and 'inhibitory', which can be interpreted as proponents and opponents for test examples that are correctly classified, and the reverse if they are misclassified. The distinction arises because the representer approach explains the prediction score and not the loss.*

## 3.2 First-order Approximation to Idealized Influence, and Extension to Minibatches

Since the step-sizes used in updating the parameters in the training process are typically quite small, we can approximate the change in the loss of a test example in a given iteration $t$ via a simple first-order approximation: $\ell(w_{t+1}, z') = \ell(w_t, z') + \nabla\ell(w_t, z') \cdot (w_{t+1} - w_t) + O(\|w_{t+1} - w_t\|^2)$. Here, the gradient is with respect to the parameters and is evaluated at $w_t$. Now, if stochastic gradient descent is utilized in training the model, using the training point $z_t$ at iteration $t$, then the change in parameters is $w_{t+1} - w_t = -\eta_t \nabla\ell(w_t, z_t)$, where $\eta_t$ is the step size in iteration $t$. Note that this formula should be changed appropriately if other optimization methods (such as AdaGrad, Adam, or Newton's method) are used to update the parameters. The first-order approximation remains valid, however, as long as a small step-size is used in the update.

For the rest of this section we restrict to gradient descent for concreteness. Substituting the change in parameters formula in the first-order approximation, and ignoring the higher-order term (which is of the order of $O(\eta_t^2)$), we arrive at the following first-order approximation for the change in the loss $\ell(w_t, z') - \ell(w_{t+1}, z') \approx \eta_t \nabla\ell(w_t, z') \cdot \nabla\ell(w_t, z_t)$. For a particular training example $z$, we can approximate the idealized influence by summing up this approximation in all the iterations in which $z$ was used to update the parameters. We call this first-order approximation $\texttt{TracIn}$, our primary notion of influence: $\texttt{TracIn}(z, z') = \sum_{t:\ z_t=z} \eta_t \nabla\ell(w_t, z') \cdot \nabla\ell(w_t, z)$.

To handle minibatches of size $b \geq 1$, we compute the influence of a minibatch on the test point $z'$, mimicking the derivation in Section 3.1, and then take its first-order approximation: First-Order Approximation$(B_t, z') = \frac{1}{b} \sum_{z \in B_t} \eta_t \nabla\ell(w_t, z') \cdot \nabla\ell(w_t, z)$, because the gradient for the minibatch $B_t$ is $\frac{1}{b} \sum_{z \in B_t} \nabla\ell(w_t, z)$. Then, for each training point $z \in B_t$, we attribute the $\frac{1}{b} \cdot \eta_t \nabla\ell(w_t, z') \cdot \nabla\ell(w_t, z)$ portion of the influence of $B_t$ on the test point $z'$. Summing up over all iterations $t$ in which a particular training point $z$ was chosen in $B_t$, we arrive at the following definition of $\texttt{TracIn}$ with minibatches: $\texttt{TracIn}(z, z') = \frac{1}{b} \sum_{t:\ z \in B_t} \eta_t \nabla\ell(w_t, z') \cdot \nabla\ell(w_t, z)$.

**Remark 3.3** *The derivation suggests a way to measure the goodness of the approximation for a given step: We can check that the change in loss for a step $\ell(w_t, z') - \ell(w_{t+1}, z')$ is approximately equal to First-Order Approximation$(B_t, z')$.*

## 3.3 Practical Heuristic Influence via Checkpoints

The method described so far does not scale to typically used long training processes since it involves tracing of the parameters, as well as training points used, at each iteration: effectively, in order to compute the influence, we need to replay the training process, which is obviously impractical. In order to make the method practical, we employ the following heuristic. It is common to store checkpoints (i.e. the current parameters) during the training process at regular intervals. Suppose we have $k$ checkpoints $w_{t_1}, w_{t_2}, \ldots, w_{t_k}$ corresponding to iterations $t_1, t_2, \ldots, t_k$. We assume that between checkpoints each training example is visited exactly once. (This assumption is only needed for an approximate version of Lemma 3.1; even without this, $\texttt{TracInCP}$ is a useful measure of influence.) Furthermore, we assume that the step size is kept constant between checkpoints, and we use the notation $\eta_i$ to denote the step size used between checkpoints $i - 1$ and $i$. While the first-order approximation of the influence needs the parameter vector at the specific iteration where a given training example is visited, since we don't have access to the parameter vector, we simply approximate it with the first checkpoint parameter vector after it. Thus, this heuristic results in the

following formula[5]:

$$\texttt{TracInCP}(z, z') = \sum_{i=1}^{k} \eta_i \nabla \ell(w_{t_i}, z) \cdot \nabla \ell(w_{t_i}, z') \tag{1}$$

**Remark 3.4 (Handling Variations of Training)** *In our derivation of `TracIn` we have assumed a certain form of training. In practice, there are likely to be differences in optimizers, learning rate schedules, the handling of minibatches etc. It should be possible to redo the derivation of `TracIn` to handle these differences. Also, we expect the practical form of `TracInCP` to remain the same across these variations.*

**Remark 3.5 (Counterfactual Interpretation)** *An alternative interpretation of Equation 1 is that it approximates the influence of a training example on a test example* had it been visited *at each of the input checkpoints. Under this counterfactual interpretation, it is valid to study the influence of a point that is not part of the training data set, keeping in mind that such an example did not impact the training process or the checkpoints that arose as a consequence of the training process.*

## 4    Evaluations

In this section we compare `TracIn` with influence functions [2] and the representer point selection method [3]. Brief descriptions of these two methods can be found in the supplementary material. We also compare practical implementations of `TracIn` against an idealized version.

### 4.1    Evaluation Approach

We use an evaluation technique that has been used by the previous papers on the topic (see Section 4.1 [3] and Section 5.4 of [2]).[6] The idea is to measure self-influence, i.e., the influence of a training point on its own loss, i.e., the training point $z$ and the test point $z'$ in `TracIn` are identical.

Incorrectly labelled examples are likely to be strong proponents (recall terminology in Section 3.1) for themselves. Strong, because they are outliers, and proponents because they would tend to reduce loss (with respect to the incorrect label). Therefore, when we sort training examples by decreasing self-influence, an effective influence computation method would tend to rank mislabelled examples in the beginning of the ranking. We use the fraction of mislabelled data recovered for different prefixes of the rank order as our evaluation metric; higher is better. (In our evaluation, we know which examples are mislabelled because we introduced them. If this technique was used to find mislabelled examples in practice, we assume that a human would inspect the list and identify mislabelling.)

To simulate the real world mislabelling errors, we first trained a model on correct data. Then, for 10% of the training data, we changed the label to the highest scoring incorrect label. We then attempt to identify mislabelled examples as discussed above.

### 4.2    CIFAR-10

In this section, we work with ResNet-56 [16] trained on the CIFAR-10 [17]. The model on the original dataset has 93.4% test accuracy. [7]

**Identifying Mislabelled Examples**    Recall the evaluation set up and metric in Section 4.1.[8]

For influence functions, it is prohibitively expensive to compute the Hessian for the whole model, so we work with parameters in the last layer, essentially considering the layers below the last as frozen.

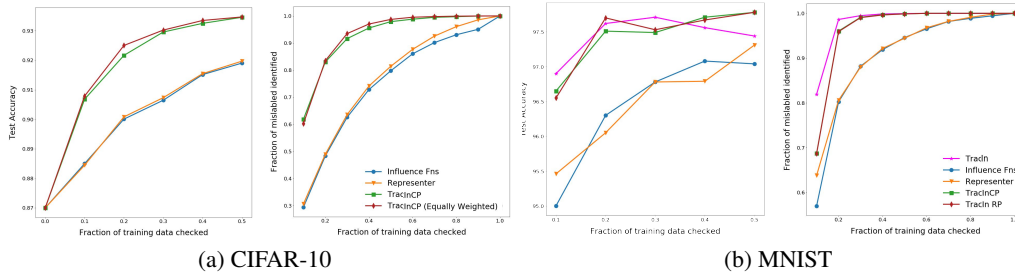

(a) CIFAR-10  (b) MNIST

Figure 1: CIFAR-10 and MNIST Mislabelled Data Identification with `TracIn` Representer points, and Influence Functions. We use "Fraction of mislabelled identified" on the y axis to compare the effectiveness of each method. (RP = Random Projections, CP = CheckPoints)

This mimics the set up in Section 5.1 of [2]. Given that CIFAR-10 only has 50K training examples, we directly compute inverse hessian by definition.

For representer points, we fine-tuned the last layer with line-search, which requires the full batch to find the stationary point and use $|\alpha_{ij}|$ as described in Section 4.1 of [3] to compare with self-influence.

We use `TracInCP` with only the last layer. We sample every 30 checkpoints starting from the 30th checkpoint; every checkpoint was at a epoch boundary. The right hand side of Figure 1a shows that `TracInCP` identifies a larger fraction of the mislabelled training data (y-axis) regardless of the fraction of the training data set that is examined (x-axis). For instance, `TracIn` recovers more than 80% of the mislabelled data in the first 20% of the ranking, whereas the other methods recover less than 50% at the same point. Furthermore, we show that *fixing* the mislabelled data found within a certain fraction of the training data, results in a larger improvement in test accuracy for `TracIn` compared to the other methods (see the plot on the left hand side of Figure 1a). We also show that weighting checkpoints equally yields similar results. This provides support to ignore learning rate for implementation simplification.

**Effect of different checkpoints on `TracIn` scores**   Next, we discuss the contributions of the different checkpoints to the scores produced by `TracIn`; recall that `TracIn` computes a weighted average across checkpoints (see the defintion of `TracInCP`). We find that different checkpoints contain different information. We identify the number of mislabelled examples from each class (the true class, not the mislabelled class) within the first 10% of the training data in Figure 8 (in the supplementary material). We show results for the 30th, 150th and 270th checkpoint. We find that the mix of classes is different between the checkpoints. The 30th checkpoint has a larger fraction (and absolute number) of mislabelled deer and frogs, while the 150th emphasizes trucks. This is likely because the model learns to identify different classes at different points in training process, *highlighting the importance of sampling checkpoints.*

## 4.3   MNIST

In this section, we work on the MNIST digit classification task[9]. Because the model is smaller than the Resnet model we used for CIFAR-10, we can perform a slightly different set of comparisons. First, we are able to compute approximate influence for each training step (Section 3.2), and not just heuristic influence using checkpoints. Second, we can apply `TracIn` and the influence functions method to all the model parameters, not just the last layer.

Since we have a large number of parameters, we resort to a randomized sketching based estimator of the influence whose description can be found in the supplementary material. In our experiments, this model would sometimes not converge, and there was significant noise in the influence scores, which are estimating a tiny effect of excluding one training point at a time. To mitigate these issues, we pick lower learning rates, and use larger batches to reduce variance, making the method time-intensive.

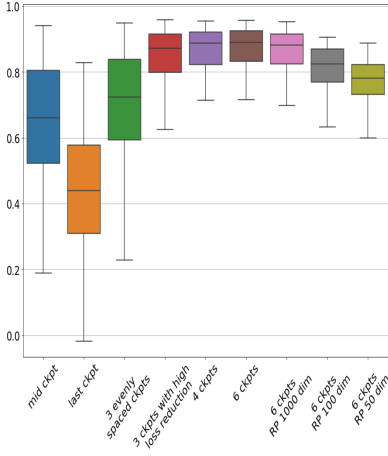

Figure 2: Analysis of effect of approximations with Pearson correlation of first order approximate `TracIn` influences with heuristic influences over multiple checkpoints and with projections of different sizes. RP stands for random projection.

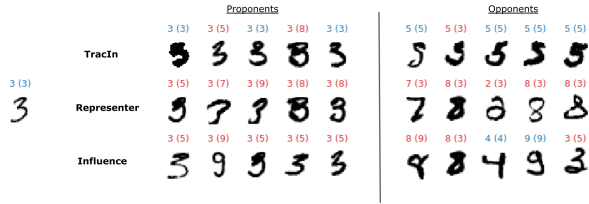

(a) Correctly classified 3.

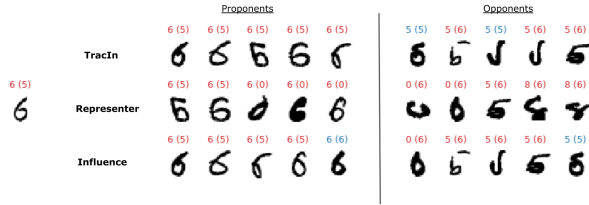

(b) Incorrectly classified 6

Figure 3: Proponents and opponents examples using `TracIn`, representer point, and influence functions. (Predicted class in brackets)

**Visual inspection of Proponents and Opponents.** We eyeball proponents and opponents of a random sample of test images from MNIST test set. We observe that `TracInCP` and representer consistently find proponents visually similar to test examples. Although, the opponents picked by representer are not always visually similar to test example (the opponent '7' in Figure 3a and '5' and '8's in Figure 3b). In contrast, `TracInCP` seems to pick pixel-wise similar opponents.

**Identifying mislabelled examples.** Recall the evaluation set up and metric in Section 4.1. We train on MNIST mislabelled data as described there.[10] Similar to CIFAR-10, `TracIn` outperforms the other two methods and retrieves a larger fraction of mislabelled examples for different fractions of training data inspected (Figure 1b). Furthermore, as expected, approximate `TracIn` is able to recover mislabelled examples faster than heuristic `TracInCP` (we use every 30th checkpoint, starting from 20th checkpoint), but not by a large margin.

Next, we evaluate the effects of our various approximations.[11]

**Effect of the First-Order Approximation.** We now evaluate the effect of the first-order approximation (described in 3.2). By Remark 3.3, we would like the total first-order influence at a step First-Order Approximate Influence$(B_t, z')$ to approximate the change in loss at the step $\ell(w_t, z') - \ell(w_t, z')$. Figure 7 (in the supplementary material) shows the relationship between the two quantities; every point corresponds to one parameter update step for one test point. We consider 100 random test points. The overall Pearson correlation between the two quantities is 0.978, which is sufficiently high.

**Effect of checkpoints.** We now discuss the approximation from Section 3.3, i.e., the effect of using checkpoints. We compute the correlation of the influence scores of 100 test points using `TracInCP` with different checkpoints against the scores from the first-order approximation `TracIn`. As discussed in Remark D (in the supplementary material), we find that selecting checkpoints with high loss reduction, are more informational than selecting same number of evenly spaced checkpoints. This is because in later checkpoints the loss flattens, hence, the loss gradients are small. Figure 2 shows `TracInCP` with just one checkpoint from middle correlates more than the last checkpoint with

`TracIn` scores. Consequently, `TracInCP` with more checkpoints improves the correlation, more so if the checkpoints picked had high loss reduction rates.

## 4.4 Discussion

We now discuss conceptual differences between `TracIn` and the other two methods.

**Influence functions:** Influence functions mimic the process of tracing the change in the loss of a test point when you *drop* an individual training point and retrain. In contrast, as discussed in Section 3, `TracIn` explains the change in the loss of a test point between the start of training and the end of training. The former counterfactual is inferior when the training data set contains copies or near copies; deleting one of the copies is likely to have no effect even though the copies together are indeed influential. Also, it is prohibitively expensive drop a datapoint and retrain the model, influence functions approximate this by using the first and second-order optimality conditions. Modern deep learning models are rarely, if ever, trained to even moderate-precision convergence, and optimality can rarely be relied upon. In contrast, `TracIn` does not need to rely optimality conditions.

**Representer Point method:** This technique computes the influence of training point using the representer theorem, which posits that when only the top layer of a neural network is trained with $\ell_2$ regularization, the obtained model parameters can be specified as a linear combination of the post-activation values of the training points at the last layer. Like the influence function approach, it too relies on *optimality conditions*. Also, it can only explain the prediction of the test point, and not its loss.

**Implementation Complexity:** Both techniques are complex to implement. The influence technique requires the the inversion of a Hessian matrix that has a size that is quadratic in the number of model parameters, and the representer point method requires a complex, memory-intensive line search or the use of a second order solver such as LBFGS. `TracIn`, in contrast, has a simple implementation.

## 5 Applications

We apply `TracIn` to a regression problem (Section 5.1) a text problem (Section 5.2) and an computer vision problem (Section 5.3) to demonstrate its ability to generate insights. This section is not meant to be an evaluation. The last of these use cases is on a ResNet-50 model trained on the (large) Imagenet dataset, demonstrating that `TracIn` scales.

Table 1: Opponents for text classification on DBPedia. All examples shown have the same label and prediction. Proponents can be found in Appendix.

| | | |
|---|---|---|
| Example | OfficeHolder | **Manuel Azaña** Manuel Azaña Díaz (Alcalá de Henares January 10 1880 – Montauban November 3 1940) was the first Prime Minister of the Second Spanish Republic (1931–1933) and later served again as Prime Minister (1936) and then as the second and last President of the Republic (1936–1939). The Spanish Civil War broke out while he was President. With the defeat of the Republic in 1939 he fled to France resigned his office and died in exile shortly afterwards. |
| Opponents | Artist | **Mikołaj Rej** Mikołaj Rej or Mikołaj Rey of Nagłowice (February 4 1505 – between September 8 and October 5 1569) was a Polish poet and prose writer of the emerging Renaissance in Poland as it succeeded the Middle Ages as well as a politician and musician. He was the first Polish author to write exclusively in the Polish language and is considered (with Biernat of Lublin and Jan Kochanowski) to be one of the founders of Polish literary language and literature. |
| Opponents | Artist | **Justin Jeffre** Justin Paul Jeffre (born on February 25 1973) is an American pop singer and politician. A long-time resident and vocal supporter of Cincinnati Jeffre is probably best known as a member of the multi-platinum selling boy band 98 Degrees.Before shooting to super stardom Jeffre was a student at the School for Creative and Performing Arts in Cincinnati. It was there that he first became friends with Nick Lachey. The two would later team up with Drew Lachey and Jeff Timmons to form 98 Degrees. |
| Opponents | Artist | **David Kitt** David Kitt (born 1975 in Dublin Ireland) is an Irish musician. He is the son of Irish politician Tom Kitt.He has released six studio albums to date: Small Moments The Big Romance Square 1 The Black and Red Notebook Not Fade Away and The Nightsaver. |

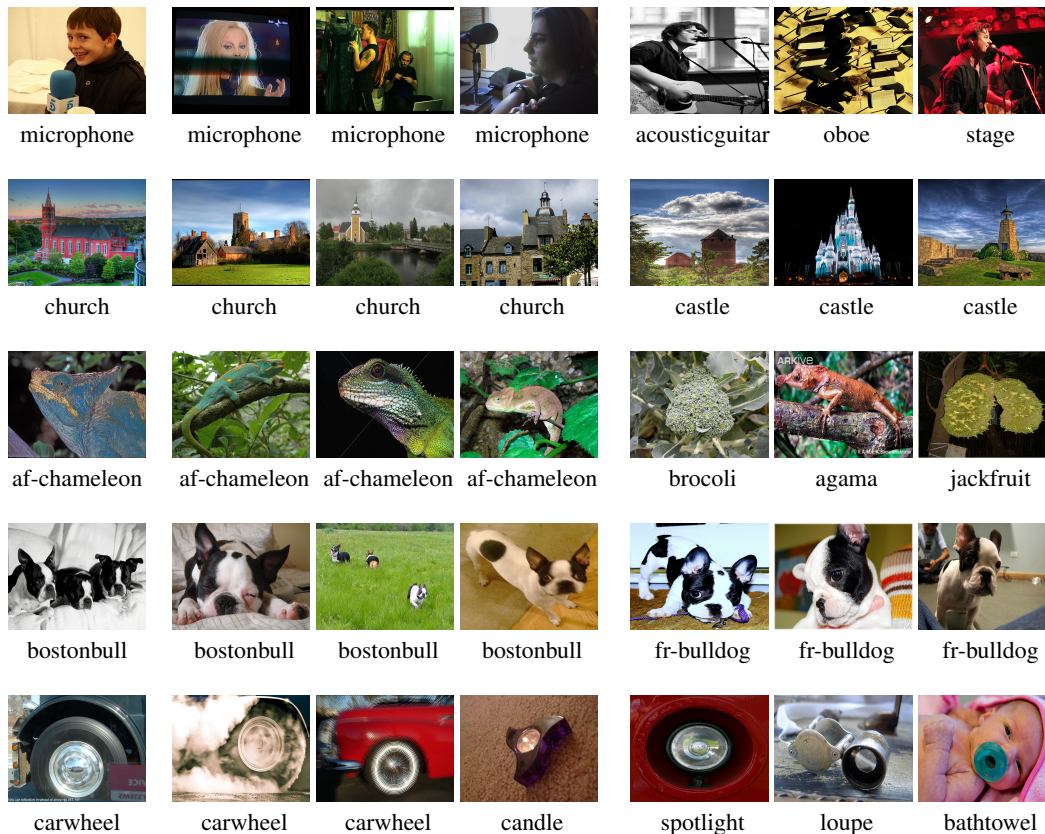

| | | | | | | |
|---|---|---|---|---|---|---|
| microphone | microphone | microphone | microphone | acousticguitar | oboe | stage |
| church | church | church | church | castle | castle | castle |
| af-chameleon | af-chameleon | af-chameleon | af-chameleon | brocoli | agama | jackfruit |
| bostonbull | bostonbull | bostonbull | bostonbull | fr-bulldog | fr-bulldog | fr-bulldog |
| carwheel | carwheel | carwheel | candle | spotlight | loupe | bathtowel |

Figure 4: `TracIn` applied on Imagenet. Each row starts with the test example followed by three proponents and three opponents. The test image in the first row is classfied as band-aid and is the only misclassified example. (af-chameleon: african-chameleon, fr-bulldog: french-bulldog)

## 5.1 California Housing Prices

We study `TracIn` on a regression problem using California housing prices dataset [18]. [12]

The notion of comparables in real estate refers to recently sold houses that are similar to a home in location, size, condition and features, and are therefore indicative of the home's market value. We can use `TracInCP` to identify model-based comparables, by examining the proponents for certain predictions. For instance, we could study proponents for houses in the city of Palo Alto, a city in the Bay Area known for expensive housing. We find that the proponents are drawn from other areas in the Bay Area, and the cities of Sacramento, San Francisco and Los Angeles. One of the influential examples lies on the island of Santa Catalina, also known for expensive housing.

We also study self-influences of training examples (see Section 4.1 for the definition of self-influence). High self-influence is more likely to be indicative of memorization. We find that the high self influence examples come from densely populated locations, where memorization is reasonable, and conversely, low self-influence ones comes from sparsely populated areas,where memorization would hurt model performance. Housing plots with geo coordinates can be found in Figure 10 (in the supplementary material).

## 5.2 Text Classification

We apply `TracIn` on the DBPedia ontology dataset introduced in [19]. The task is to predict the ontology with title and abstract from Wikipedia. The dataset consists of 560K training examples and 70K test examples equally distributed among 14 classes. We train a Simple Word-Embedding Model (SWEM) [20] for 60 epochs and use the default parameters of sentencepeice library as tokenizer [21] and achieve 95.5% on both training and test. We apply `TracInCP`and sample 6 evenly spaced checkpoints and the gradients are taken with respect to the last fully connected layer.

Table 1 shows the top 3 opponents for one test example (Manuel Azana); we filter misclassified training examples from the list to find a clearer pattern. (Misclassified examples have high loss, and therefore high training loss gradient, and are strong proponents/opponents for different test examples, and are thus not very disciminative.) The list of opponents provide insight about data introducing correlation between politicians and artists.

## 5.3 Imagenet Classification

We apply `TracIn` on the fully connected layer of ResNet-50 trained on Imagenet [22][13] We use a trick to reduce dimensionality: It relies on the fact that for fully-connected layers, the gradient of the loss w.r.t. the weights for the layer is a rank 1 matrix. Thus, `TracIn` involves computing the dot (Hadamard) product of two rank 1 matrices. Details are in the supplementary material.

We show three proponents and three opponents for five examples in figure 4. We filtered out misclassified examples as we did for text classification. A few quick observations: (i) The proponents are mostly images from the same label. (ii) In the first row of figure 4, the style of the microphone in the test example is different from the top proponents, perhaps augmenting the data with more images that resemble the test one can fix the misclassification. (iii) For the correctly classified test examples, the opponents give us an idea which examples would confuse the model (for the church, there are castles, for the bostonbull there are french bulldogs, for the wheel there are loupes and spotlights, and for the chameleon there is a closely related animal (agama) but there are also broccoli and jackfruits.

# 6 Conclusion

In this paper we propose a method called `TracIn` to identify the influence of a training data point on a test point.

The method is *simple*, a feature that distinguishes it from previously proposed methods. Implementing `TracIn` only requires only a basic understanding of standard machine learning concepts like gradients, checkpoints and loss functions.

The method is *general*. It applies to any machine learning model trained using stochastic gradient descent or a variant of it, agnostic of architecture, domain and task.

The method is *versatile*. The notion of influence can be used to explain a single prediction (see Section 5), or identify mislabelled examples (see Section 4). Over time, we expect other applications to emerge. For instance, [23] uses it to perform a kind of active learning, i.e., to expand a handful of hard examples into a larger set of hard examples to fortify toxic speech classifiers.

Finally, we note that some human judgment is required to apply `TracIn` correctly. To feed it reasonable inputs, i.e., checkpoints, layers, and loss heads. To interpret the output correctly; for instance, to inspect sufficiently many influential examples so as to account for the loss on the test example. Lastly, as with any statistical measure (e.g. mutual information Pearson correlation etc.), we need to ensure that the measure is meaningfully utilized within a broader context.

# 7 Acknowledgements

We would like to thank the anonymous reviewers, Qiqi Yan, Binbin Xiong, and Salem Haykal for discussions.

## 8 Broader Impact

This paper proposes a practical technique to understand the influence of training data points on predictions. For most real world applications, the impact of improving the quality of training data is simply to improve the quality of the model. In this sense, we expect the broader impact to be positive.

For models that impact humans, for instance a loan application model, the technique could be used to examine the connection between biased training data and biased predictions. Again, we expect the societal impact to be generally positive. However there is an odd chance that an inaccuracy in our method results in calling a model fair when it is not, or unfair, when it is actually fair. This potential negative impact is amplified in the hands of an adversary looking to prove a point, one way or the other. It is also possible that the technique could be used to identify modifications to the training data that hurt predictions broadly or for some narrow category.

## Footnotes

[3]With the minor difference that the training process is a discrete process, whereas the path used within the fundamental theorem is continuous.

[4]By test example, we simply mean an example whose prediction is being explained. It doesn't have to be in the test set.

[5]In a sense, the checkpoint based approximation is an improvement on the idealized definition of `TracIn` because it ignores the order in which the training data points were visited by the specific training run; this sequence will change for a different run.

[6]This is just an evaluation approach. There are possibly other ways to detect mislabelled examples (e.g. [15]) that don't use the notion of training data influence/attribution.

[7]All model for CIFAR-10 are trained with 270 epochs with a batch size of 1000 and 0.1 as initial learning rate and the following schedule (1.0, 15), (0.1, 90), (0.01, 180), (0.001, 240) where we apply learning rate warm up in the first 15 epochs.

[8]Training on the mislabelled data reduces test accuracy from 93.4% to 87.0% (train accuracy is 99.6%).

[9]We use a model with 3 hidden layers and  240K parameters. This model has 97.55% test accuracy.

[10]After 140 epochs, it achieves accuracy of 89.94% on mislabelled train set, and 89.95% on test set.

[11]We use the same 3-layer model architecture, but with the correct MNIST dataset. The model has 97.55% test set accuracy on test set, and 99.30% train accuracy.

[12]We used a 8:2 train-test split and trained a regression model with 3 hidden layers with 168K parameters, using Adam optimizer minimizing MSE for 200 epochs. The model achieves explained variance of 0.70 on test set, and 0.72 on train set. We use every 20th checkpoint to get `TracIn` influences.

[13]The model is trained for 90 epochs and achieves 73% top-1 accuracy. The training data consists of 1.28M training examples with 1000 classes. The 30th, 60th, and 90th checkpoints are used for `TracInCP` and we project the gradients to a vector of size 1472.

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
