[Supplementary Material]

# A Description of Influence Functions and Representer Point Methods

## A.1 Influence Functions

[2] proposed using the idea of Influence functions [24] to measure the influence of a training point on a test example. Specifically, they use optimality conditions for the model parameters to mimic the effect of perturbing single training example:

$$\text{Inf}(z, z') = -\nabla_{\hat{w}}\ell(\hat{w}, z') \cdot H_{\hat{w}}^{-1} \cdot \nabla_{\hat{w}}\ell(\hat{w}, z). \tag{2}$$

Here, $H_{\hat{w}} = \frac{1}{n}\sum_1^n \nabla^2\ell(\hat{w}, z_i)$ is the Hessian. As pointed out by [2], for large deep learning models with massive training sets, the inverse Hessian computation is costly and complex. This technique also assumes that the model is at convergence so that the optimality conditions hold.

**Scalable implementation via randomized sketching** It becomes infeasible to compute the inverse Hessian when the number of parameters is very large, as is common in modern deep learning models. To mitigate this issue we compute randomized estimators of $H_{\hat{w}}^{-1}\nabla_{\hat{w}}\ell(\hat{w}, z')$ via a *sketch* of the inverse Hessian in the form of the product $H_{\hat{w}}^{-1}G^\top$ where $G$ is the same kind of random matrix as in Section E. The product $[H_{\hat{w}}^{-1}G^\top][G\nabla_{\hat{w}}\ell(\hat{w}, z')]$ is then an unbiased estimator of $H_{\hat{w}}^{-1}\nabla_{\hat{w}}\ell(\hat{w}, z')$. Note that the sketch takes only $O(dp)$ memory rather than $O(p^2)$ that the inverse Hessian would take. We compute the sketch by solving the optimization problem $\min_S \|H_{\hat{w}}S - G^\top\|_F^2$, via a customized stochastic gradient descent procedure based on the formula

$$\nabla_S \|H_{\hat{w}}S - G^\top\|_F^2 = 2H_{\hat{w}}(H_{\hat{w}}S - G^\top).$$

This customized stochastic gradient descent procedure uses the following stochastic gradient computed using *two* indepdently chosen minibatches of examples $B_1, B_2$ instead of the customary one:

$$2[\tfrac{1}{|B_1|}\textstyle\sum_{z\in B_1} \nabla^2\ell(\hat{w}, z)][\tfrac{1}{|B_2|}\textstyle\sum_{z\in B_2} \nabla^2\ell(\hat{w}, z)S - G^\top]. \tag{3}$$

Note that $\mathbb{E}[\frac{1}{B_1}\sum_{z\in B_1} \nabla^2\ell(\hat{w}, z)] = H_{\hat{w}}$ and $\mathbb{E}[\frac{1}{|B_2|}\sum_{z\in B_2} \nabla^2\ell(\hat{w}, z)] = H_{\hat{w}}$, and since $B_1$ and $B_2$ are independently chosen, we conclude that the expectation of the quantity in (3) is indeed $2H_{\hat{w}}(H_{\hat{w}}S - G^\top)$ as required. Note that (3) can be computed using Hessian-vector products, which can be computed easily using the Pearlmutter trick [25].

## A.2 Representer Point Selection

The second method is proposed in [3] and is based on the representer point theorem [26]. The method decomposes the logits for any test point into a weighted combination of dot products between the representation of the test point at the top layer of a neural network and those of the training points; this is effectively a kernel method. The weights in the decomposition capture the influences of that training points.

Specifically, consider a neural network model with fitted parameters into $\{w_1, w_2\}$, where $w_2$ is the matrix of parameters that produces the logits from the input representation (i.e. the top layer weights) and $w_1$ are the remaining parameters. To meet the conditions of the representer theorem, the final layer of the model is tuned by adding a term L2 regularization term $\lambda \|w_2\|^2$ to the loss and training the model to convergence. This optimization produces a new set of parameters $w_2'$ for the last layer, resulting in a new model with parameters $w' = \{w_1, w_2'\}$. Then the influence of a training example $z$ on a test example $z'$ is a $k$-dimensional vector (one element per class) given by

$$\text{Rep}(z, z') =$$
$$-\frac{1}{2\lambda n}(f(w_1, z) \cdot f(w_1, z'))\partial_{\phi(w', z')}\ell(w', z'). \tag{4}$$

Here, $f(w_1, z)$ is the input representation, i.e. the outputs of the last hidden layer, and $\phi(w', z) = w_2'f(w_1, z)$ are the logits. The L2 regularization requires a complex, memory-intensive line search, and results in a model different from the original one, possibly resulting in influences that are

Figure 5: CIFAR-10 results: Proponents and opponents examples of a correctly classified cat for influence functions, representer point, and `TracIn`. (Predicted class in brackets)

Figure 6: CIFAR-10 results: Proponents and opponents examples of an incorrectly classified automobile for influence functions, representer point, and `TracIn`. (Predicted class in brackets)

unfaithful to the original model. Conceptually, it is also not clear how to study the influence that flows via the parameters in lower layers—computing a stationary point is harder in this situation. Furthermore, both the influence functions approach and `TracIn` could be used to explain the influence of a training example on the loss of a test example or its prediction score. In contrast, it is unclear how to use the representer point method to explain loss on a test example.

## B  A Visual Inspection of Proponents and Opponents for CIFAR

We now consider the same training procedure in Section 4.1 but on the regular CIFAR-10 dataset. We show the top 5 proponent and opponent examples of an image from the test set and compare the three methods qualitatively in Figures 5 and 6. All three methods retrieved mostly cats as positive examples and dogs as negative examples, but `TracIn` seems more consistent on the types of cats and dogs. For the mis-classified automobile, proponents of `TracIn` pick up automobiles of a similar variety type.

## C  Low Latency Implementation

We can use an approximate nearest neighbors technique to quickly identify influential examples for a specific prediction. The idea is to pre-compute the training loss gradients at the various checkpoints (possibly using the random projection trick to reduce space, see Section E in appendix for details). Then, we concatenate the loss gradients for a given training point $z$ (i.e., $\ell(w_{t_1}, z), \ell(w_{t_2}, z) \ldots \ell(w_{t_k}, z)$) together into one vector. This can be then loaded into an approximate nearest neighbor library(e.g. [27]). During analysis, we can do the same for a test example—the gradient calls for the different checkpoints can be done in parallel. We then invoke nearest neighbor search. The nearest neighbor library then performs the computation implicit in `TracInCP`.

## D    Selecting Checkpoints

In the application of `TracInCP`, we choose checkpoints at epoch boundaries, i.e., between checkpoints, each training example is visited exactly once. However, it is possible to be smarter about how checkpoints are chosen: Generally, it makes sense to sample checkpoints at points in the training process where there is a steady decrease in loss, and to sample more frequently when the rate of decrease is higher. It is worth avoiding checkpoints at the beginning of training when loss fluctuates. Also, checkpoints that are selected after training has converged add little to the result, because the loss gradients here are tiny. Relatedly, computing `TracInCP` with *just* the final model could result in noisy results.

## E    Random Projection Approximation

Modern deep learning models frequently have a huge number of parameters, making the inner product computations in the first-order approximation of the influence expensive, especially in the case where the influence on a number of different test points needs to be computed. In this situation, we can speed up the computations significantly by using the the technique of random projections. This method allows us to pre-compute low-memory sketches of the loss gradients of the training points which can then be used to compute randomized unbiased estimators of the influence on a given test point. The same sketches can be re-used for multiple test points, leading to computational savings. This is a well-known technique (see for example [28]) and here we give a brief description of how this is done. Choose a random matrix $G \in \mathbb{R}^{d \times p}$, where $d \ll p$ is a user-defined dimension for the random projections (larger $d$ leads to lower variance in the estimators), whose entries are sampled i.i.d. from $\mathcal{N}(0, \frac{1}{d})$, so that $\mathbb{E}[G^\top G] = I$. We compute the following sketch: in iteration $t$, compute and save $\eta_t G \nabla \ell(w_t, z_t)$. Then given a test point $z'$, the dot product $(\eta_t G \nabla \ell(w_t, z_t)) \cdot (G \nabla \ell(w_t, z'))$ is an unbiased estimator of $\eta_t \nabla \ell(w_t, z_t)) \cdot (\nabla \ell(w_t, z'))$, and can thus be substituted in all influence computations.

## F    Fast Random Projections for Gradients of Fully-Connected Layers

Suppose we have a fully connected layer in the neural network with a weight matrix $W \in \mathbb{R}^{m \times n}$, where $m$ is the number of units in the input to that layer, and the $n$ is the number of units in the output of the layer. For the purpose of `TracIn` computations, it is possible to obtain a random projection of the gradient w.r.t. $W$ into $d$ dimensions with time and space complexity $O((m+n) \cdot \sqrt{d})$ rather than the naive $O(mnd)$ complexity that the standard random projection needs.

To formalize this, let us represent the layer as performing the following computation: $y := Wx$ where $x \in \mathbb{R}^n$ is the input to the layer, and $y$ is the vector of pre-activations (i.e. the value fed into the activation function). Now suppose we want to compute the gradient of some function $f$ (e.g. loss, or prediction score) of the output of the layer, i.e. we want to compute $\nabla_W(f(Wx))$. A simple application of the chain rule shows gives the following formula for the gradient:

$$\nabla_W(f(Wx)) = \nabla_y f(y) x^\top.$$

In particular, note that the gradient w.r.t. $W$ is rank 1. This property is very useful for `TracIn` since it involves computations of the form $\nabla_W(f(Wx)) \cdot \nabla_W(f'(Wx'))$, where $f'$ is another function and $x'$ is another input. Note that for $y' = Wx'$, we have

$$\nabla_W(f(Wx)) \cdot \nabla_W(f'(Wx'))$$
$$= (\nabla_y f(y) x^\top) \cdot (\nabla_{y'} f'(y') {x'}^\top)$$
$$= (\nabla_y f(y) \cdot \nabla_{y'} f'(y'))(x \cdot x').$$

The final expression can be computed in $O(m+n)$ time by computing the two dot products $(\nabla_y f(y) \cdot \nabla_{y'} f'(y'))$ and $(x \cdot x')$ separately and then multiplying them. This is much faster than the naive dot product of the gradients, which takes $O(mn)$ time.

This can already speed up `TracIn` computations. We can also save on space by randomly projecting $\nabla_y f(y)$ and $x$ separately, but unfortunately this doesn't seem to be amenable to fast nearest-neighbor search. If we want to use fast nearest-neighbor search, we will need to use random projections in

the following manner which also exploits the rank-1 property. To project into $d$ dimensions, we can use two independently chosen random projection matrices $G_1 \in \mathbb{R}^{\sqrt{d} \times m}$ and $G_2 \in \mathbb{R}^{\sqrt{d} \times n}$, with $\mathbb{E}[G_1 G_1^\top] = \mathbb{E}[G_2 G_2^\top] = I$, and compute

$$G_1 \nabla_y f(y) x^\top G_2^\top \in \mathbb{R}^{\sqrt{d} \times \sqrt{d}},$$

which can be flattened to a $d$-dimensional vector. Note that this computation requires time and space complexity $O((m+n) \cdot \sqrt{d})$. Furthermore, since $G_1$ and $G_2$ are chosen independently, it is easy to check that

$$\mathbb{E}[(G_1 \nabla_y f(y) x^\top G_2^\top) \cdot (G_1 \nabla_{y'} f'(y') x'^\top G_2^\top)]$$
$$= (\nabla_y f(y) x^\top) \cdot (\nabla_{y'} f'(y') x'^\top),$$

so the randomized dot-product is unbiased.

# G  Additional Results

This section contains charts and images that support discussions in the main body of the paper.

Figure 7: Comparison of change in loss at all training steps and `TracIn` influences at those steps for 100 test examples from MNIST dataset. This measures the quality of the first-order approximation–see Section 4.3.

Figure 8: Number of identified mislabelled examples by class for three checkpoints within the top 10% of ranking by self-influence. Different checkpoints highlight different labels—see Section 4.2.

Proponents

| | 1 (8) | 1 (8) | 1 (8) | 1 (1) | 1 (4) | 8 (1) | 8 (1) | 8 (8) | 9 (9) | 8 (1) |

**TracIn**

1 (1)

| | 1 (8) | 1 (8) | 1 (3) | 1 (4) | 1 (8) | 4 (1) | 2 (1) | 7 (1) | 4 (1) | 8 (1) |

**Representer**

| | 2 (3) | 1 (8) | 3 (5) | 2 (3) | 6 (5) | 4 (2) | 9 (4) | 5 (6) | 4 (4) | 6 (6) |

**Influence**

Opponents

(a)

Proponents

| | 4 (4) | 4 (4) | 4 (4) | 4 (4) | 4 (9) | 9 (9) | 9 (4) | 9 (9) | 9 (4) | 9 (8) |

**TracIn**

4 (4)

| | 4 (1) | 4 (1) | 4 (9) | 4 (9) | 4 (8) | 7 (4) | 9 (4) | 1 (4) | 9 (4) | 7 (4) |

**Representer**

| | 4 (4) | 4 (4) | 7 (9) | 4 (4) | 4 (4) | 9 (9) | 6 (5) | 9 (4) | 9 (4) | 9 (4) |

**Influence**

Opponents

(b)

Proponents

| | 7 (7) | 7 (7) | 7 (7) | 7 (7) | 7 (7) | 9 (7) | 9 (8) | 9 (9) | 8 (8) | 9 (7) |

**TracIn**

7 (7)

| | 7 (1) | 7 (2) | 7 (3) | 7 (2) | 7 (4) | 1 (7) | 3 (7) | 9 (7) | 4 (7) | 3 (7) |

**Representer**

| | 7 (9) | 7 (4) | 7 (7) | 7 (4) | 7 (1) | 2 (7) | 1 (7) | 1 (7) | 4 (4) | 9 (7) |

**Influence**

Opponents

(c)

Figure 9: MNIST(Section 4.3): Proponents and opponents examples of a correctly classified images for TracIn, representer point, and influence functions. (Predicted class in brackets).

(a) Influential training examples for 11 test examples in city of Palo Alto, with entire dataset in yellow.

(b) Training examples with high and low self influences showing dense areas which model memorizes, and sparsely populated areas where model learns from examples away from the area.

Figure 10: `TracIn` on California housing prices dataset.

Table 2: Proponents for text classification on DBPedia—see Section 5.2. All examples shown have the same label and prediction.

| | | |
|---|---|---|
| Example | OfficeHolder | **Manuel Azaña** Manuel Azaña Díaz (Alcalá de Henares January 10 1880 – Montauban November 3 1940) was the first Prime Minister of the Second Spanish Republic (1931–1933) and later served again as Prime Minister (1936) and then as the second and last President of the Republic (1936–1939). The Spanish Civil War broke out while he was President. With the defeat of the Republic in 1939 he fled to France resigned his office and died in exile shortly afterwards. |
| Proponents | OfficeHolder | **Annemarie Huber-Hotz** Annemarie Huber-Hotz (born 16 August 1948 in Baar Zug) was Federal Chancellor of Switzerland between 2000 and 2007. She was nominated by the FDP for the office and elected on 15 December 1999 after four rounds of voting. The activity is comparable to an office for Minister. The Federal Chancellery with about 180 workers performs administrative functions relating to the co-ordination of the Swiss Federal government and the work of the Swiss Federal Council. |
| Proponents | OfficeHolder | **José Manuel Restrepo Vélez** José Manuel Restrepo Vélez (30 December 1781 – 1 April 1863) was an investigator of Colombian flora political figure and historian. The Orchid genus Restrepia was named in his honor.Restrepo was born in the town of Envigado Antioquia in the Colombian Mid-west. He graduated as a lawyer from the Colegio de San Bartolomé in the city of Santa Fe de Bogotá. He later worked as Secretary for Juan del Corral and Governor Dionisio Tejada during their dictatorial government over Antioquia. |
| Proponents | OfficeHolder | **K. C. Chan** Professor Ceajer Ka-keung Chan (Traditional Chinese: 陳家強) SBS JP (born 1957) also referred as KC Chan is the Secretary for Financial Services and the Treasury in the Government of Hong Kong. He is also the ex officio chairman of the Kowloon-Canton Railway Corporation and an ex officio member of the Hong Kong International Theme Parks Board of Directors. |