[Reviews · NeurIPS 2020]

Review 1

Summary and Contributions: This paper introduces a method for identifying the influence of a certain training sample in training neural networks using gradient-based optimization. The idea is to use first order approximation of actual influence of this training sample on a testing sample. The authors further introduced an approach to implement this idea in an actual training process. That is, to use the checkpoints saved during training as sparse replay for the full training process. The method is able to identify samples that contributes to increase or decrease of testing loss values. The authors evaluated the approach on both small-scale datasets and large-scale application datasets (like ImageNet). The quantitative results show that in controlled experiments the method is better in finding mislabelled samples. In qualitative analysis on ImageNet, the method is able to identify visually meaningful training samples that leads to confusion or mis-classification on testing samples.

Strengths: The problem of understanding the dynamics of training neural networks is important. Understanding the influence of each individual sample in the training is one of the approaches towards this goal. This paper provides a viable way towards. This merits it attention from the community. + The technical description is sound. The idea is first order approximation is neat and well motivated. + The proposed solution is novel as far as I am aware of. It shows better efficiency in computation and better effectiveness in characterizing the influence of training samples. + The experimental evaluation is adequate. + A good portion of bias in machine learning comes from training data. This works tackles the problem of measuring training samples' influence in an efficient way. I believe this will also help industrial practitioners adopt and use it in fighting bias in machine learning models.

Weaknesses: The experiment of fixing mis-labelled samples are related to other approaches towards training with noisy labels. It is beneficial to compare the effectiveness using the TrackIn to do this against these methods. This will provide some guidance to practitioners. Other than fixing wrong labels, the influence measured by the current method is not very easily assessed. This makes it difficult to compare between methods working on this problem, even though it is understandably important. I would expect a solid quantitative measurement to be presented.

Correctness: The claims and method are correct to my understanding. The empirical methodology is sound.

Clarity: Yes. The flow is easy to follow. The major message is clearly conveyed.

Relation to Prior Work: Yes.

Reproducibility: No

Additional Feedback: Please see the weakness section for suggestions. Overall I feel this is a good idea and well executed. It tackles an important problem with a novel solution. I would vote for its acceptance. ============= Added after author feedback: After reading the authors' feedback, I think some of my concerns are addressed. This is a nice work on an important problem. I would like to keep my initial rating. But I would still recommend the authors to think about the quantitative measurement on how well the influence is discovered, which will help the community to extend its own influence.


Review 2

Summary and Contributions: The paper addresses and analyzes the quality of the training data by proposing a method to quantify the influence of the training samples on the test samples. The TrackIn method defines and efficiently computes a value for this relation (influence), validating the observations through empirical evaluation on CIFAR and MNIST. Compared with other approaches, this method does not rely on the model optimality.

Strengths: The proposed solution is based on a simple observation of expressing the influence between a (train, test) data points pair using the differences in loss before and after updating the model using the train sample. The authors managed to scale this observation by approximating it in checkpoints. Further, they measure the quality of the approach by checking the influence of a point on its own loss (correlation). The method is simple, but novel, having clear advantages over previous solutions: the self-influence score is better it does not rely on the optimality It is computationally efficient The relevance to the community is proven in the Applications section were the authors apply their method on several larger and diverse datasets and models, extracting some insights for training data and proving the power, the validity and usefulness of the method (NLP: DBPedia, CV: Imagenet + resnet50, house price regression). The method can be important from the Bias Reduction and Explainability point of view (as mentioned în the Broader Impact section).

Weaknesses: The experimental setup involves introducing an artificial percent of mislabeled samples. Is the performance of the method influenced by choosing a different percent? How? The evaluation metric takes into account the recall for the mislabeled examples (in top sorted samples by self-influence). In a non-toy dataset or in one with less wrong labels, it would be difficult to use this solution to cherry pick by hand mislabeled samples. Reporting also the precision here would be helpful to know where we are from this perspective.

Correctness: The performed experiments, the theory and the approximations seem to be just right.

Clarity: Overall, the paper is well organized and easy to read. The experiments (subtitles) for MNIST and CIFAR have a different structure and I think would be easier to follow if they would look in the same manner (“Identifying mislabelled examples.”, “Effect of checkpoints” sections should have the same names).

Relation to Prior Work: The prior work similarities and differences were introduced at the beginning and frequently referenced through the paper in places where punctual comparisons could be done. Also, they were discussed from the evaluation point of view.

Reproducibility: Yes

Additional Feedback: How do the authors explain the difference between the classes ratio for mislabeled examples in different checkpoints? More precisely, why at the end of training (last checkpoint) all classes have a similar number of mislabeled examples in top 10% (why truck class detection significantly decreases in the last checkpoint). Typos: “it is prohibitively expensive drop a datapoint” “The influence technique requires the the inversion” “TrackInCPand” ======================================== After reviewing authors' response, I maintain my rating and support this as a solid and important work.


Review 3

Summary and Contributions: Paper proposes a method called TrackIn which computes the influence of a training example on a test example prediction, by tracking the changes in the training loss. Paper compares their proposed method on mislabelled training examples identification task with influence functions and representer points, and claim that their proposed method is better than both influence functions and representer points. Authors also emphasize that their proposed method is computationally efficient than influence functions and representer points.

Strengths: Understanding the influence of training data on model prediction is an important problem to understand the behavior of complex deep learning models. Influence functions and representer point selection are two main techniques to measure the influence. This paper proposes a very simple yet effective technique to estimate the influence. Unlike previous methods, the proposed technique is easy to implement and is very scalable. The proposed TrackIn method is described in detail. Paper provides detailed analysis on mislabelled examples identification on the image classification task, and also provide ablation studies to support approximations used such as First-Order Approximation, the importance of using multiple checkpoints.

Weaknesses: I have some major concerns with the evaluation part of the paper. 1. Paper compared their method with influence functions and representer selection. A simple baseline could be a loss based selection method. Simply select training points based on loss change. A recent paper [DataLens IJCNN 20] shows that a simple loss based selection outperforms both influence functions and representer selection on mislabelled data identification when the mislabeled data is small. As the fraction of mislabelled data increases, influence function works better than loss based method. 2. Paper doesn't show the performance of TrackIn with varying amounts of mislabelled data. As pointed above, I expect TrackIn to perform poorly when we increase the mislabelled data. 3. Checkpoint ensembling is a widely used technique in machine translation [MT ensemble WMT16], semi-supervised learning [Temporal Ensembling ICLR17], knowledge distillation [KD Distillation NAACL 19] so it's not surprising that it helps in TrackIn. One can argue that influence functions can also benefit from the checkpoint ensembling. The authors should explain that. Also, the paper should cite prior work related to checkpoint ensembling as a motivation for picking multiple checkpoints.

Correctness: Presented claims and method are correct. I have a few concerns related to evaluation protocol which I have raised in the weaknesses section.

Clarity: Yes the paper is well written and is easy to understand.

Relation to Prior Work: Paper mainly focuses on 3 prior papers including influence functions, representer points and Data cleansing with SGD. Authors clearly describe these techniques and provide pros and cons of TrackIn vs influence functions/representer points.

Reproducibility: Yes

Additional Feedback: ### Update after reading author response ### I would like to thanks authors for answering my questions/concerns. I think it's important to show how TrackIn is gonna behave when we increase the noise ratio. Such experiments will help reader in understanding the strengths/weaknesses of your proposed approach. Please refer to weaknesses section for suggestions/questions. References: [Temporal Ensembling ICLR17] Temporal Ensembling for Semi-Supervised Learning [MT ensemble WMT16] Edinburgh Neural Machine Translation Systems for WMT 16 [KD Distillation NAACL 19] Online Distilling from Checkpoints for Neural Machine Translation [DataLens IJCNN 20] Interpreting Deep Models through the Lens of Data

[Author Response · NeurIPS 2020]

# Author Response

We thank the reviewers for their valuable feedback.

**R1: Comparison to "training with noisy labels".**
**R3: Comparison to "loss based selection method" from [DataLens IJCNN 20].**

We were simply following an evaluation technique proposed by the two previous papers (influence functions, representer) on the topic. In this sense, identifying mislabeled examples using self-influence is simply a way to compare *influence techniques*. We do not claim to be the best way to fix or work with mislabelled data.

**R1: Other than fixing wrong labels, the influence measured by the current method is not very easily assessed.**

It is more challenging empirically to evaluate influence techniques (which *depend on how the model operates) in comparison to prediction/classification problems (where ground truth is specified independent of the model). Besides the "fixing labels" eval, we also provide conceptual arguments in favor or our method (Appendix: Section A), and comparative visual results on CIFAR (Appendix: Figure 6) and MNIST (Appendix: Figure 9).

**R2: The experimental setup involves introducing an artificial percent of mislabeled samples. Is the performance of the method influenced by choosing a different percent?**
**R3: I expect TrackIn to perform poorly when we increase the mislabelled data.**

Yes, one would expect *any self-influence based technique to perform poorly when the fraction of mislabelled data is high (say >30%). But this does not imply that TrackIn would do worse than representer or influence functions.

That said, we picked what we thought was a practically reasonable rate of mislabeling.

**R2: In a non-toy dataset or in one with less wrong labels, it would be difficult to use this solution to cherry pick by hand mislabeled samples. Reporting also the precision here would be helpful to know where we are from this perspective.**

The goal of the evaluation with a fixed percentage of mislabeled examples is to compare with prior works which also use the same metric. The trend should be the same regardless of precision or recall. We agree that reporting precision would be helpful in a "non-toy" dataset with less wrong labels and we will make this point in our next revision.

**R2: How do the authors explain the difference between the classes ratio for mislabeled examples in different checkpoints?... at the end of training all classes have a similar number of mislabeled examples in top 10.**

During the training process, the decrease in loss for each class (averaged over instances of the class) is not uniform. Frogs and Deers converge pretty early, and then Trucks. Therefore, for earlier checkpoints, the self-influence technique is more effective on these classes. In the final checkpoint, the model has converged to 99% accuracy, i.e., it is doing well on all classes, consequently, the performance of the self-influence technique is similar across classes.

**R3: Checkpoint ensembling is a widely used technique One can argue that influence functions can also benefit from the checkpoint ensembling. Also, the paper should cite prior work related to checkpoint ensembling as a motivation for picking multiple checkpoints.**

Notice that for us checkpoint ensembling *arises* from trying to practically implement the mathematical form of Idealized TrackIn (Lemma 3.1); in this sense our motivation for using checkpoints is perhaps different. We will definitely cite the suggested literature to point out the resemblance.

While other influence techniques may also benefit from checkpoint ensembling, they remain harder to implement than TrackIn.

[Meta-Review · NeurIPS 2020]

I recommend accepting this paper. The authors managed to address the concerns that the reviewers had. Everyone is in agreement that this paper should be accepted.